# Molecular Pathogenesis of Pulmonary Fibrosis, with Focus on Pathways Related to TGF-β and the Ubiquitin-Proteasome Pathway

**DOI:** 10.3390/ijms22116107

**Published:** 2021-06-05

**Authors:** Naoki Inui, Satoshi Sakai, Masatoshi Kitagawa

**Affiliations:** 1Department of Clinical Pharmacology and Therapeutics, Hamamatsu University School of Medicine, 1-20-1 Handayama, Higashi-ku, Hamamatsu, Shizuoka 431-3192, Japan; 2Department of Molecular Biology, Hamamatsu University School of Medicine, 1-20-1 Handayama, Higashi-ku, Hamamatsu, Shizuoka 431-3192, Japan; ssakai@hama-med.ac.jp

**Keywords:** E3 ligase, endothelial cell, epithelial mesenchymal transition (EMT), fibroblast, idiopathic pulmonary fibrosis (IPF), TGF-β, ubiquitin proteasome system

## Abstract

Idiopathic pulmonary fibrosis (IPF) is a progressive and fatal interstitial lung disease. During the past decade, novel pathogenic mechanisms of IPF have been elucidated that have shifted the concept of IPF from an inflammatory-driven to an epithelial-driven disease. Dysregulated repair responses induced by recurrent epithelial cell damage and excessive extracellular matrix accumulation result in pulmonary fibrosis. Although there is currently no curative therapy for IPF, two medications, pirfenidone and nintedanib, have been introduced based on understanding the pathogenesis of the disease. In this review, we discuss advances in understanding IPF pathogenesis, highlighting epithelial–mesenchymal transition (EMT), the ubiquitin-proteasome system, and endothelial cells. TGF-β is a central regulator involved in EMT and pulmonary fibrosis. HECT-, RING finger-, and U-box-type E3 ubiquitin ligases regulate TGF-β-Smad pathway-mediated EMT via the ubiquitin-proteasome pathway. p27 degradation mediated by the SCF-type E3 ligase, Skp2, contributes to the progression of pulmonary fibrosis by promotion of either mesenchymal fibroblast proliferation, EMT, or both. In addition to fibroblasts as key effector cells in myofibroblast differentiation and extracellular matrix deposition, endothelial cells also play a role in the processes of IPF. Endothelial cells can transform into myofibroblasts; therefore, endothelial–mesenchymal transition can be another source of myofibroblasts.

## 1. Introduction

Interstitial lung disease (ILD) is a diverse group of respiratory diseases with varying pathology. ILD and pulmonary fibrosis, an advanced pathological condition, are characterized by lung parenchyma injury with various patterns of interstitial inflammation, cell proliferation, and fibrosis within the alveolar wall and interstitium [1,2,3]. The pathogenesis of ILDs is not yet fully understood, although hundreds of etiological factors are estimated to be involved [1,2]. Cases in which the cause cannot be identified are called idiopathic interstitial pneumonia.

Idiopathic pulmonary fibrosis (IPF) is the most common form of idiopathic interstitial pneumonia and idiopathic fibrotic lung disorder. IPF occurs primarily in middle-aged and older adults, typically in the sixth and seventh decades, and its incidence increases remarkably with age [4,5,6]. Increased age in combination with interstitial findings on high-resolution computed tomography (HRCT) of the chest are strong predictors of IPF [7]. More men than women have IPF, with a sex ratio of 7:3 [4,5,6,8]. Hutchinson et al. estimated a conservative incidence range of 3–9 cases per 100,000 for Europe and North America [9]. In an epidemiological survey of Japanese patients, the cumulative incidence and prevalence of IPF were 2.2 and 10.0 per 100,000, respectively [10].

Fibrosis results in impaired ventilation, functional respiratory impairment, reduced gas exchange, and respiratory failure [1]. IPF is characterized by a chronic, progressive, irreversible, and usually lethal clinical course [1,2,3,4,5,6,11]. Most patients have respiratory symptoms, such as coughing and dyspnea on exertion, and develop worsening pulmonary function and show progressive fibrosis on HRCT over years [4,5]. IPF patients have diverse and unpredictable clinical courses and the median survival is 2.5–3.5 years after diagnosis [4,5,6,8]. Unfortunately, the condition of most patients worsens in the years following diagnosis and they die from acute exacerbation, progression of chronic respiratory failure, and lung cancer [4,5,8].

The diagnostic criteria for idiopathic interstitial pneumonias and IPF have changed several times [1,5,12], and those for IPF were revised recently [12] to emphasize the significance of either radiological, histopathological, or in combination, features or patterns of usual interstitial pneumonia (UIP) [6,12]. Chest HRCT is the imaging test of choice and is regarded as an essential examination for a diagnosis of IPF [2,4,5,12]. A UIP pattern on HRCT images is the hallmark radiological pattern of IPF and has a high predictive value for the histopathological pattern of UIP [2,8,12]. It is distributed predominantly in the subpleural and basal locations. Subpleural clustering of cystic airspaces, called “honeycombing”, is a feature of the UIP pattern and is critical for achieving a definite diagnosis [8]. The presence of reticular opacities and traction bronchiectasis is common. Subpleural, basal-predominant reticular abnormalities with peripheral traction bronchiectasis without honeycombing are regarded as a probable UIP pattern on HRCT images [2,8,12]. If all UIP criteria are not met on HRCT images, a lung biopsy may be recommended [2,12], and a UIP pattern on histopathological examination is a hallmark of IPF [4,5,8,12]. The key features of a UIP pattern include architectural destruction, fibrosis with scarring and honeycomb changes, and scattered fibroblastic foci (Figure 1A,B) [1,4,8]. Involvements show heterogeneous patchy distribution within the periphery of the acinus or lobule dominance and a lack of uniform involvement in the parenchyma [3,11,13], which is an essential diagnostic criterion. These histopathological changes predominate in the subpleural and paraseptal parenchyma. Inflammatory changes, consisting of patchy interstitial infiltration of lymphocytes and plasma cells with hyperplasic type II pneumocytes, are mild to moderate [4,12]. The fibrotic involvement mainly consists of dense collagen and fibroblast foci. Smooth muscle metaplasia in the interstitium is seen in areas of fibrosis and honeycomb change. In the diagnostic process, emphasis should be placed on multidisciplinary discussions between pulmonologists, radiologists, and pathologists experienced in the diagnosis of ILD [4,5,8,12,13].

There is currently no curative medication for IPF [5]. In past decades, anti-inflammatory agents, corticosteroids, and immunosuppressive agents, were used for IPF treatment, based on the hypothesis that chronic inflammation in the alveolar compartment was the primary mechanism of pulmonary fibrosis [4,8,14,15]. These medications did not improve survival or pulmonary function [2], and prednisone, azathioprine, and N-acetylcysteine combination therapy adversely increased mortality and hospitalization [16]. Now, corticosteroid therapy is not recommended for patients with IPF [2,5,12]. Recent progress in understanding the pathogenesis of IPF, which involves many mediators and signaling pathways, has led to the development of new IPF drugs. Two such new medications, pirfenidone and nintedanib, effectively slowed disease progression in placebo-controlled randomized phase 3 studies [8,14]. They reduced the decline in lung function among IPF patients but did not improve survival or quality of life. Pirfenidone is an oral synthetic pyridine derivative with broad anti-fibrotic and anti-inflammatory properties. Although the mechanisms by which the fibrotic process can be modified are poorly understood, fibrosis is slowed down by modulating procollagen transcription and limiting the activation and differentiation of fibroblasts into myofibroblasts [17]. Analysis of pooled data from three phase 3 trials showed that pirfenidone reduced the risk of death at one year by 48% compared with a placebo (hazard ratio: 0.52, 95% CI: 0.31–0.87) and reduced the decline in walking distance during a 6-min walk [18]. Nintedanib is a small molecule orally-bioavailable indolinone derivative tyrosine kinase inhibitor that inhibits multiple signaling pathways involving the vascular endothelial growth factor (VEGF) receptor, fibroblast growth factor (FGF) receptor, and platelet-derived growth factor (PDGF) receptor [8]. In an international placebo-controlled double-blind clinical phase 3 trial, nintedanib reduced the annual rate of decline in forced vital capacity [19]. However, no data are available to guide which agent should be administered as the first-choice therapy [2,8]. Aside from pirfenidone and nintedanib, many other targeted therapies are now also in early-phase clinical trials [11].

## 2. Overview of IPF Pathogenesis and Risk Factors

### 2.1. Current Concepts of IPF Pathogenesis

There are numerous studies demonstrating that inflammation can be quite pronounced in IPF pathogenesis and, formerly, inflammation was regarded as the main cause of fibrosis [4,8,14]. Anti-inflammatory agents, such as steroids and immune suppressants, have previously been used to treat IPF, but accumulated data indicate that inflammation is not causal in many patients with IPF [2,16,20]. The current hypothesis for IPF pathogenesis and pathophysiology suggests that recurrent epithelial cell damage initiates and induces a dysregulated repair response, and excessive accumulation of extracellular matrix secreted by myofibroblasts results in pulmonary fibrosis (Figure 2) [2,4,11,13,14,19,21]. Unusual functions and phenotypes of epithelial cells, alveolar macrophages, T cells, and fibroblasts are assumed to contribute to pulmonary fibrogenesis [20]. King et al. described how the concept of IPF pathogenesis has shifted from an inflammatory-driven to an epithelial-driven disease [4]. The primary site of injury is the interstitium, the space between the epithelial and endothelial basement membranes [1]. In normal circumstances, the initial injury would be repaired and the tissue would be reconstructed, but in IPF impaired healing induces abnormal epithelial–mesenchymal interactions [13]. Impaired wound healing involves a disordered, epithelial-dependent, fibroblast-activated process of reactivation of developmental signaling pathways resulting in the secretion of a cascade of proinflammatory molecules that activate fibroblast proliferation, and myofibroblast induction and deposition [4,13].

A variety of profibrotic mediators, such as TGF-β, PDGF, and connective tissue growth factor (CTGF), and their signaling cascades play important roles in the pathogenesis of fibrotic diseases [22,23]. During the initial injury phase, a variety of cells in the lungs, including alveolar macrophages, epithelial cells, inflammatory cells, fibroblasts, and myofibroblasts, produce and release a diverse set of potent fibrogenic mediators [15,24,25]. These mediators are involved in epithelial cell injury and apoptosis. In addition, they directly recruit and activate fibroblasts to produce collagen [13].

### 2.2. TGF-β in IPF

TGF-β is a dimeric polypeptide growth factor. TGF-β regulates cell proliferation and differentiation, immune regulation, embryonic development, and angiogenesis [14,26]. In the wound healing process, TGF-β is released from alveolar epithelial cells, macrophages, platelet granules, and infiltrating regulatory T cells after an injury event and promotes wound repair by increasing the production and deposition of extracellular matrix, inflammatory cell recruitment, and fibroblast differentiation [14].

TGF-β is critical in the initiation and progression of all types of tissue fibrosis (Figure 2) [27] and its overproduction leads to excessive deposition of scar and tissue fibrosis [14]. Increased TGF-β expression occurs in patients with fibrotic kidney diseases, hepatic fibrosis, systemic sclerosis, and pulmonary fibrosis. In the process of fibrosis, TGF-β, as the most potent profibrotic mediator, recruits and activates monocytes, circulating fibrocytes, and fibroblasts [14,26]. TGF-β promotes gene transcription to increase the production of collagen, fibronectin, and proteoglycans, thereby elevating the production and deposition of extracellular matrix [3,14,26]. In addition, TGF-β reduces the function of enzymes involved in extracellular matrix degradation, such as collagenase, heparinase, matrix metalloproteinases, plasminogen activators, elastases, and stromelysin, and increases the activity of enzymes that inhibit the degradation of the extracellular matrix, including plasminogen activator inhibitor type 1 (PAI-1) and tissue inhibitor of metalloprotease [3,26]. Epithelial, hematopoietic, neuronal, and connective tissue cells can produce TGF-β and have TGF-β receptors (TβRs) [26].

### 2.3. Other Profibrotic Mediators in IPF

The TGF-β system is complex and interacts with other cell-signaling pathways. TGF-β promotes the production of profibrotic mediators and other proangiogenic mediators. In addition to TGF-β, a variety of other secreted factors, including tumor necrosis factor-α (TNF-α) [28], CTGF [29], PDGF, and FGF, are involved in expansion of the fibroblast population, myofibroblast differentiation, and extracellular matrix accumulation (Figure 2) [14]. CTGF is a small peptide that has a variety of stimulatory effects on fibroblasts [14]. CTGF mediates TGF-β-induced collagen synthesis by fibroblasts. Blockade of CTGF synthesis or action inhibits collagen synthesis and fibroblast accumulation, which reduces TGF-β-induced tissue formation [30]. PDGF is a common growth factor produced in many cells, including macrophages, platelets, endothelial cells, and fibroblasts [14]. PDGF is upregulated by TGF-β stimulation [14], is a potent mitogen for fibroblasts, and plays an essential role in the expansion of myofibroblasts [23]. Overproduction of PDGF can induce heart, liver, and renal fibrosis. FGF regulates cell proliferation, differentiation, migration, and survival [23]. FGF-2 is a potent mitogen for fibroblasts, airway smooth muscle cells, and type II alveolar epithelial cells [31], and induces collagen synthesis in lung fibroblasts and myofibroblasts [24].

Regarding interleukins, levels of the proinflammatory cytokine, IL-18, are elevated in serum, airway epithelium, and alveolar macrophages of patients with IPF [32]. Hoshino et al. demonstrated that bleomycin increased caspase-1–dependent IL-1β and IL-18 expression in mice [33]. The NLRP3 inflammasome, an intracellular multimeric protein complex, is critical for host immune defenses and mediates caspase-1 activation and the secretion of IL-1β and IL-18 [34]. Recently, the NLRP3 inflammasome has become of special interest in occupational pulmonary fibrosis and IPF [35,36]. The NLRP3 inflammasome is activated in alveolar epithelial cells in tissue fibrosis, which induces myofibroblast differentiation of lung-resident mesenchymal stem cells [37]. Jäger et al. showed that the NLRP3-inflammasome-caspase 1 pathway is hyper-inducible in bronchoalveolar lavage fluid cells from IPF and especially during acute exacerbation [38].

### 2.4. Risk Factors in IPF

In fibrotic diseases, unidentified triggers alter inflammatory and fibrotic responses. However, these triggers are not specific and the development and progression of IPF depend on interactions with environmental factors and genetic disease susceptibility [8,23]. Repeated microinjury of the alveolar epithelium, caused by an interaction between a genetic predisposition to abnormal epithelial cell regulation and injurious environmental agents [14,23], has been recognized as the first step in the pathogenesis of IPF [20]. Environmental factors, cigarette smoking, mineral and wood dust, oxidative stress, microaspiration, and viral infection can be related to epithelial cell damage and pulmonary fibrosis development and progression [2,4,5,11,13,14]. Smoking is strongly associated with IPF [4,5,8]. Most patients have a history of cigarette smoking and smokers with IPF have a shorter survival than those who have never smoked [20]. Song et al. showed the relationship between smoking and endoplasmic reticulum stress in lung fibroblast–myofibroblast differentiation using cigarette smoke extract [39]. Gastroesophageal reflux disease is common in patients with IPF and is regarded as another microenvironmental factor [40]. Although an association between IPF and gastroesophageal reflux has been proposed, its pathophysiological mechanisms and the usefulness of antacid therapy remain largely unknown [8]. Viruses such as Epstein–Barr virus, human herpes viruses 7 and 8, cytomegalovirus, hepatitis C, herpes simplex virus, parvovirus B19, and torque teno virus have also been implicated in the pathogenesis of IPF [20,22].

Age-related changes, a genetic susceptibility, epigenetic factors, and mitochondrial dysfunction determine disease susceptibility [4,41]. Epigenetic changes include alterations in DNA methylation, activation of the unfolded protein response in response to endoplasmic reticulum stress, histone tail modification, or microRNA expression [4,11,15,42]. Senescence or aging of epithelial cells and fibroblasts appear to be among the factors promoting pulmonary fibrosis [2,43]. Genome-wide association studies have identified several genetic variants associated with the development of IPF that can be considered IPF susceptibility genes, such as telomerase reverse transcriptase (*TERT*), desmoplakin (*DSP*), and A-kinase anchoring protein 13 (*AKAP13*) [44]. A common polymorphism in the promoter region of the gene encoding airway mucin 5B (*MUC5B*) is found in pulmonary fibrosis patients [4,8,14,20,38]. MUC5B is involved in airway clearance and bacterial host defense [6]. The minor allele of the single nucleotide polymorphism (SNP) rs35705950, located in the promoter region, induces *MUC5B* overexpression in airway epithelial cells. This minor allele was present at a frequency of 38% in patients with IPF. The odds ratios for disease among patients who were heterozygous and those who were homozygous for the minor allele of this SNP were 9.0 (95% CI: 6.2–13.1) and 21.8 (95% CI: 5.1–93.5), respectively [45]. However, these genetic mutations do not induce pulmonary fibrosis directly in mouse models [11]. Wolter et al. suggested that genetic mutations may not be sufficient to cause pulmonary fibrosis and additional environmental exposure may be required for the development of pulmonary fibrosis.

## 3. Animal Models of Pulmonary Fibrosis

Studies using animal models enable the pathogenesis of diseases to be interrogated and facilitate preclinical assessment of drug candidates [2,23]. For pulmonary fibrosis, histological assessment of collagen accumulation by measuring hydroxyproline is recommended by the American Thoracic Society panel [46]. Bleomycin is the most widely used agent to induce and reproduce pulmonary fibrosis in experimental animals [13,47]. Bleomycin can be administered by inhalation, intratracheal, subcutaneous, intraperitoneal, and transvenous routes [46]. Systemic bleomycin administration directly injures pulmonary endothelial cells [48] and, subsequently, epithelial cell injury, inflammation, and fibrosis occurs [47]. Airway administration of bleomycin is the most frequently used route and initially injures airway epithelial cells. Administration can be a single or repeated doses. A repeat administration regimen can reproduce chronic injury with more fibrotic changes [46]. Intratracheal administration can stimulate lung injury and resultant fibrosis with a single bleomycin dose. The American Thoracic Society panel considers the murine intratracheal bleomycin model the best characterized animal model available for preclinical testing [48]. Bleomycin induces injury that develops chronologically: interstitial inflammation, fibroproliferation, and extracellular matrix protein deposition. This offers opportunities to study cell–cell interactions and the soluble mediators that induce pathological fibrotic changes [20,47]. There are two morphological phases in pulmonary fibrosis formation in animal models [47,49]. In the intratracheal model, the inflammation predominant phase can be observed within one to two weeks. Patchy inflammatory cell infiltration in the alveolar walls and epithelial injury with reactive hyperplasia are pathologically observed. This is followed by the fibrotic phase, where maximal collagen deposition and fibrotic changes are observed between the third and fourth week (Figure 1C–E) [46,49,50,51]. The effects of bleomycin are variable according to dose, route of delivery, and animal species [46,49]. The morphological fibrotic changes resolve spontaneously 28 days after bleomycin administration, which is a major limitation of the bleomycin model [13,49]. No model can completely reproduce all the physiological and histopathological features of human pulmonary fibrosis (Figure 1) [23,46,49], although the mouse models can help to elucidate the mechanism of this complex disease.

## 4. Importance of Fibroblasts in the Pathogenesis of Interstitial Pneumonia

Fibroblasts from patients with IPF exhibit an abnormally “activated” phenotype and have global alterations in DNA methylation, which may contribute to fibroblast heterogeneity among patients with IPF [52]. In IPF, fibroblasts are recruited, activated, and induced to transdifferentiate to myofibroblasts (Figure 2). Activated fibroblasts secrete collagen, elastin, glycoproteins, proteoglycans, and profibrotic mediators that undergo pathological remodeling and abnormal crosslinking, which changes the mechanical properties of the pulmonary extracellular matrix [14,31]. Rockey et al. suggested that activated lung fibroblasts may cause alveolar cell apoptosis, which leads to further fibroblast activation, injury, and effector cell activation [27]. IPF fibroblasts seem to have increased resistance to apoptosis [31]. In normal wound healing, unrequired fibroblasts are removed by clearance mechanism with the activation of apoptosis to limit excess matrix deposition and fibrosis [11], although the factors that differentiate normal wound repair from fibrosis are unknown [16]. Elimination of myofibroblasts by apoptosis does not occur in the fibroblastic foci of IPF [4]. Impaired activity of these processes can cause pathological scarring and fibrosis.

Fibroblastic foci are regions of highly proliferative myofibroblast accumulation that are located immediately adjacent to regions of hyperplastic or apoptotic epithelial cells [13]. Myofibroblasts secrete large amounts of extracellular matrix molecules, leading to further development of pulmonary remodeling and fibrosis along with decreased removal of extracellular matrix (Figure 2) [11].

There has been impressive progress in understanding the pathogenesis of IPF. In the later part of this review, we focus on advances regarding the ubiquitin-proteasome system and endothelial cells involved in the development of pulmonary fibrosis.

## 5. Endothelial Cells in the Pathogenesis of Interstitial Pneumonia

### 5.1. Importance of Endothelial Cells in the Pathogenesis of Interstitial Pneumonia

Morphologically, pulmonary endothelial cells are contiguous to epithelial and mesenchymal cells and form the endothelial lining of blood vessels [53]. The endothelial cells control vasomotor tone, angiogenesis, leukocyte trafficking, innate and acquired immunity, hemostatic balance, and permeability. Their phenotypic, structural, and functional plasticity is affected by components of the microenvironment [53].

Although pulmonary endothelial cell injury has not been considered a main clinical topic in patients with ILDs, some clinical studies have focused on endothelial cell damage. Takabatake et al. showed pulmonary microvascular endothelial injury during a fibrotic process using the rate of ^123^I-metaiodobenzylguanidine washout from the lungs [54]. Magro et al. showed that specimens from biopsy-proven pulmonary fibrosis had morphological evidence of microvascular injury to the endothelium [55]. Endothelial cells are identified among the various cells that orchestrate and contribute to the biological processes of fibrotic changes, in addition to inflammatory cells, epithelial cells, and fibrogenic effector cells [27]. Based on the assumption that endothelial cell injury plays an important role in the processes of pulmonary fibrosis, an in vivo intratracheal bleomycin-induced pulmonary fibrosis mouse model was created as described in the previous section, and the phenotype and function of pulmonary endothelial cells was examined [50]. Mouse pulmonary endothelial cells are isolated by dissociating lung tissue into single cell suspensions using a single cell separator and the CD45–CD31^+^ cell population obtained was cultured as mouse pulmonary endothelial cells. In endothelial cells from bleomycin-treated lungs, the expression levels of endothelial injury markers, PAI-1, von Willebrand factor, and matrix metalloproteinase-12, were elevated at the early stage and sustained after a few weeks, which suggested that endothelial cells were damaged in the intratracheal bleomycin-induced pulmonary fibrosis.

Endothelium-derived relaxing factors, intracellular nitric oxide, and prostaglandin I_2_, maintain respiratory homeostasis by modulating platelet aggregation, inhibiting leukocyte adhesion, and controlling vascular smooth muscle cell proliferation [56,57]. Although nitric oxide, a gaseous free radical, protects cells from oxidant-induced injury, a high amount of nitric oxide adversely induces inflammation and apoptosis. Production of nitric oxide in endothelial cells may be involved in fibrosis [56], although the findings regarding the contribution of nitric oxide to the process of pulmonary fibrosis are inconsistent. Endothelial cells constitutively synthesize and release intracellular nitric oxide and prostaglandin I_2_ in response to agonists. Basically, endothelial cells constitutively synthesize and release intracellular nitric oxide and prostaglandin I_2_ in response to agonists [58]. The production of intracellular nitric oxide and prostaglandin I_2_ in response to thapsigargin stimulation in endothelial cells was investigated during the process of bleomycin-induced lung injury. Thapsigargin is a selective inhibitor of the endoplasmic reticulum Ca^2+^-ATPase and increases intracellular Ca^2+^ levels through plasma membrane store-operated Ca^2+^ channels. In normal conditions, the addition of thapsigargin increases the production of nitric oxide and prostaglandin I_2_. The response to the addition of thapsigargin in endothelial cells from intratracheally bleomycin-induced pulmonary fibrosis mice was maintained, but the levels of intracellular nitric oxide were reduced in endothelial cells from bleomycin-treated mice. The concentration of prostaglandin I_2_ was estimated based on that of its stable metabolite, 6-keto PGF1α. Thapsigargin simulation increased 6-keto PGF1α production in endothelial cells isolated from bleomycin-treated mice. However, the production of 6-keto PGF1α in response to thapsigargin was lower in cells isolated from bleomycin-treated mice, which showed that endothelial cell function based on thapsigargin reactivity was attenuated in endothelial cells from bleomycin-treated mice at the fibrotic phase.

Nitric oxide synthase is affected by inducible nitric oxide synthase (iNOS) and endothelial nitric oxide synthase (eNOS) expression. iNOS is stimulated by inflammation. eNOS overexpression reduced lung collagen accumulation, histological changes, and mortality in bleomycin-induced pulmonary fibrosis mice [59]. On the other hand, eNOS knockout prolonged the fibrotic changes after bleomycin exposure [60]. In endothelial cells from bleomycin-treated lungs, mRNA expression of iNOS was elevated in endothelial cells only during the acute phase.

Epithelial–mesenchymal transition (EMT) is a pathophysiological process in which epithelial cells lose some of their epithelial characteristics and gain mesenchymal ones. This molecular programming occurs in three biological contexts, development, cancer, and fibrosis, in which tissue injury and remodeling disrupt normal tissue homeostasis [11]. During EMT, cells lose epithelial markers, including E-cadherin and cytokeratins, alter surfactant production, and gain mesenchymal markers, such as N-cadherin, vimentin, smooth muscle actin, and fibronectin, and increase extracellular matrix or metalloproteinase secretion [61]. EMT is regulated by multiple extracellular ligands. Myofibroblasts are the primary effector cells in fibrotic change, defined as fibroblast-like cells expressing α-smooth muscle actin (SMA) [21,27,62]. After lung injury, the numbers of myofibroblasts increase, along with the production of collagen and other extracellular matrix components. The origin of myofibroblasts in pulmonary fibrosis is still unclear and controversial. Proposed potential sources for activated myofibroblasts include tissue-resident lung mesenchymal cells, epithelial cells, local fibroblasts, pericytes, circulating fibrocytes, and circulating and bone marrow-derived progenitor stem cells [13,27,62,63]. An attractive idea is that endothelial cells could transform to myofibroblasts and so the endothelial–mesenchymal transition could be another source of myofibroblasts [15,22,62,64]. The endothelial–mesenchymal transition changes endothelial cells into a mesenchymal phenotype, which is defined based on morphology and the potential to have both endothelial and mesenchymal markers. Through the endothelial–mesenchymal transition, endothelial cells may participate in the pathogenesis of fibrosis through secretion and deposition of excess collagen in tissues. During the endothelial–mesenchymal transition, endothelial cells lose their specific markers, such as expression of CD31 and vascular endothelial cadherin, and acquire a fibroblast-like mesenchymal phenotype, expressing α-SMA, vimentin, and type I collagen. These cells also have motile activity and migrate into surrounding tissues [62]. Hashimoto et al. showed that lesions of fibrotic changes contained significant numbers of α-SMA-stained myofibroblasts that transitioned from endothelial cells [65]. They suggested that combined treatment with TGF-β and activated Ras induced de novo α-SMA expression in microvascular endothelial cells.

### 5.2. Endothelial Cells and TGF-β

TGF-β is a central regulator of myofibroblast recruitment, activation, and differentiation during tissue repair [66] and is one of the most-studied growth factors involved in EMT [61]. mRNA and protein levels of TGF-β and TβR I were upregulated in pulmonary endothelial cells isolated from a bleomycin-induced fibrosis model. TGF-β promotes fibroblast transformation into myofibroblasts [20]. In bleomycin-induced pulmonary fibrosis mouse model, TGF-β enhanced the number of α-SMA-positive cells and the expression of Twist-1 in pulmonary endothelial cells, indicating that the endothelial cells had mesenchymal properties. TGF-β increased type I collagen transcription and the collagen content from pulmonary endothelial cells. Interestingly, TGF-β addition decreased the intracellular nitric oxide concentrations and the thapsigargin-induced increase in 6-keto PGF_1α_ in endothelial cells. Both bleomycin exposure and TGF-β substantially changed endothelial cells, functionally and phenotypically.

## 6. Regulatory Factors of TGF-β-Mediated EMT and Fibrosis

TGF-β signaling promotes induction of EMT and expression of fibrosis associated genes and participates in progression of tissue fibrosis, such as kidney and lung fibroses (Figure 3). TGF-β binds to and activates its receptor complex (TβRI and TβRII) and promotes phosphorylation-mediated activation of receptor-regulated Smads (R-Smads) Smad2/3 [67]. Activated Smad2/3 forms a complex with common partner Smads (co-Smads), Smad4, and the complex translocates into the nucleus to bind certain Smad binding sequences in the promoter regions of target genes, including fibronectin, collagens, and PAI-1, to promote their transcription. Moreover, TGF-β-signaling transduces via not only this Smad-dependent canonical pathway, but also Smad-independent non-canonical pathways such as Ras-MAPK, TAK1-p38/JNK, PI3K-Akt, and Par6-Smurf1 [68]. The Ras-ERK pathway participates in regulation of alveolar epithelial cell differentiation to α-SMA-positive myofibroblasts [69]. Various cellular proteins are involved in tissue fibrosis via regulation of the TGF-β-Smad pathway. They are inhibitory Smads (I-Smad) and Smad-cofactors [70]. I-Smad Smad7 is induced by TGF-β signaling, which binds to activated TβRI to inhibit phosphorylation of R-Smad and negatively regulate the TGF-β-Smad pathway [71]. IL-7 induces Smad7 expression via the JAK-STAT pathway and inhibits bleomycin-induced pulmonary fibrosis by inhibiting TGF-β signaling [72]. Non-coding RNAs also contribute to controlling the EMT process (Figure 3). Micro RNAs (miRNAs) target mRNAs of genes associated with TGF-β-induced EMT and pulmonary fibrosis [73]. MiR-21 inhibits translation of Smad7 as a positive regulator of TGF-β-signaling [74]. Several miRNAs target either TβRs, Smads, or both, as negative regulators of TGF-β signaling. Long non-coding RNAs (lncRNAs) such as lncRNA-ATB, TUG1, lnc-TS-1, lnc-LFAR1, and TGFB2-AS1, participate in controlling either TGF-β-mediated EMT, fibrosis, or both, via their respective molecular functions [75]. Additionally, lncRNA ELIT-1 is induced by TGF-β and binds to Smad3 to facilitate recruitment to promoters of Smad target genes such as fibronectin, PAI-1, and Snail. Thus, ELIT-1 promotes expression of EMT- and fibrosis-associated genes via the TGF-β-Smad pathway as a Smad cofactor [76] and it should be clarified whether ELIT-1 participates in pulmonary fibrosis.

## 7. Regulation of EMT and Pulmonary Fibrosis via Ubiquitin-Proteasome Pathway

### 7.1. Overview of the Ubiquitin-Proteasome Pathway

Ubiquitin-mediated protein degradation plays crucial roles in controlling the amount of cellular proteins and participates in various cellular processes such as development, cell proliferation, differentiation, signal transduction, and apoptosis. Deregulation of the system is caused by various diseases, including cancers, neurodegenerative diseases, and tissue fibrosis. E1 (ubiquitin-activating enzyme), E2 (ubiquitin-conjugating enzyme), and E3 (ubiquitin ligase) coordinately promote polyubiquitin conjugation to target proteins. Polyubiquitylated proteins are recognized and proteolytically degraded by the proteasome [77]. E3 is responsible for the specificity of ubiquitin binding to the target protein. Human cells have several hundred E3 genes. There are several types of E3 ligases, including HECT, RING finger, and U-box types. HECT-type E3 ligases, such as Smurf1/2 and Nedd4, have a HECT domain that binds to E2 and acts as a ubiquitin acceptor from E2 [78]. RING finger-type E3 ligases have a RING finger domain that binds to E2 as an active center of E3 enzymatic activity [79]. They include not only single RING E3 ligases such as Mdm2 and Cbl but also complex-type RING finger E3 ligases such as APC/cyclosome and SCF-type E3 ligases. U-box-type E3 ligases, such as CHIP and STUB1, have a U-box domain with structural similarity to the RING-finger domain [80]. Several specific E3 ligases participate in pulmonary fibrosis as indicated below (Table 1).

### 7.2. Regulation of TGF-β-Mediated EMT and Fibrosis by E3 Ubiquitin Ligases

Smurf (Smad ubiquitination regulatory factor) HECT-type E3 ligases, such as Smurf1/2, act as an accelerator to enhance TGF-β signaling by promoting ubiquitin-dependent degradation of Smad7 (Figure 3). Smurf2 is induced by TGF-β-signaling [89] and promotes pulmonary fibrosis [81] (Table 1) and obstructive kidney fibrosis [82]. miR411-3p [90] and miR-27a-3p [91] inhibit silica- and bleomycin-induced pulmonary fibrosis by suppressing Smurf2 expression.

NEDD4L (NEDD4-2) is a membrane bound HECT-type E3 ligase that targets various membrane proteins including ENaC and TβR (Figure 3). Cell type-specific deletion of NEDD4L in surfactant C-expressing type II lung epithelial cells increases the ENaC protein level and a cystic fibrosis-like disease phenotype [83]. Moreover, Duerr et al. have reported that NEDD4L suppresses bleomycin-induced pulmonary fibrosis by degradation of ENaC and TβR using NEDD4L-conditional deletion mice [84]. NEDD4L plays an important role in normal functioning of pulmonary epithelial cells, and its dysfunction causes pulmonary fibrosis (Table 1).

HECT-type E3 ligases, such as Smurfs and NEDD4L, and a single RING-type E3 Arkadia participate in modulation of fibrosis via regulation of the TGF-β-Smad pathway (Figure 3). Elkouris et al. reported that Arkadia degrades inhibitory Smad Smad7 which is methylated by Set9 and promotes bleomycin-induced pulmonary fibrosis as well as Ad-TGF-β-induced pulmonary fibrosis, whereas Smurf1/2 inhibits pulmonary fibrosis via degradation of the unmethylated Smad7-TβR complex [85] (Table 1).

U-box-type E3 ligases, such as STUB1 and CHIP, participate in ubiquitin-mediated degradation of Smads in TGF-β-signaling. NOX4 promotes TGF-β-mediated myoblast differentiation by stimulation of reactive oxygen species (ROS) induction. STUB1 targets Smad3 [92] and NOX4 for degradation to negatively regulate TGF-β-signaling [86]. STUB1-mediated degradation of NOX4 inhibits myofibroblast differentiation, which results in bleomycin-induced pulmonary fibrosis [86] (Figure 3). Because azithromycin inhibits autophagy, which activates STUB1, it may be a potential therapeutic drug for pulmonary fibrosis by promoting degradation of both NOX4 and Smad3 (Table 1).

Reports have shown that TRB3 participates in regulation of TGF-β signaling. TRB3 is a mammalian Tribble-related protein that is involved in various cellular processes as a scaffold or adaptor to promote degradation of target proteins such as Smurf1/2 and to regulate multiple signal transduction pathways such as TGF-β and MAPK signaling [93]. In mouse type II alveolar epithelial cells (MLE-12), TGF-β induces TRB3 and depletion of TRB3 decreases of pSmad3 and fibrosis-related genes such as vimentin and fibronectin [94]. Additionally, TRB3 promotes EMT through the Wnt/β-catenin signaling pathway induced by TGF-β in MLE-12 cells [95]. Moreover bleomycin-induced pulmonary fibrosis in mice is regulated by TRB3 through activation of MAPK signaling [96]. These findings suggest that TRB3 is involved in EMT and fibrosis via TGF-β-canonical and non-canonical pathways (Figure 3).

### 7.3. HSP90 Promotes EMT and IPF

Recently, small molecule inhibitors of HSP90 (Heat shock protein 90) have attracted attention as therapeutic drugs for IPF [97,98]. HSP90 is induced by various cellular stresses and participates as a molecular chaperone in the folding of client proteins. The unfolded client proteins are refolded by HSP90 collaborating with HSP40, HSP70, and cochaperones. Any residual unfolded client proteins are ubiquitylated by chaperone-dependent-E3 ligases, such as CHIP, and degraded via the ubiquitin-dependent proteasome pathway. Thus, HSP90 is involved in regulating various biological processes via the quality control of the client proteins. However, HSP90 is also associated with various diseases, such as cancers and IPF. HSP90 is required for and promotes cell proliferation and progression in various cancers and HSP90 inhibitors, such as 17AAG, AUY-922, and 17DAG, are anti-cancer drugs. In IPF patients, HSP90 is induced in the lungs [99,100]. Moreover, HSP90 is overexpressed in the lungs of lung fibrosis mouse models. HSP90 is suggested to contribute to TGF-β signaling via both canonical and non-canonical pathways in IPF [99,101]. As described above, TGF-β signaling promotes EMT of pulmonary epithelial cells to myofibroblast cells and increases the expression of fibrosis-related proteins, such as collagens and fibronectin. HSP90 participates in EMT via stabilization of TGF-β receptor complexes and in nuclear localization of Smads complexes via binding with Smad4 [102]. In a non-canonical pathway, HSP90 contributes to EMT via activation of ERK signaling [103]. As HSP90 is necessary for various biological processes as a critical factor of protein quality control, HSP90 is considered to be involved in the onset and progression of IPF by mechanisms other than the promotion of TGF-β signaling. 

Marinova et al. reported that HSP90 inhibitor, AUY-922, suppressed lung injury and fibrosis in a hydrochloric acid-induced lung injury model [104]. Solopov et al. also reported that AUY-922 suppressed nitrogen mustard-induced pulmonary fibrosis [105]. HSP90 inhibitors improve TGF-β receptor stability, canonical and non-canonical TGF-β signaling and EMT resulting in inhibition of fibrosis-associated gene expression. HSP90 inhibitors are, therefore, potential IPF therapeutics [106].

### 7.4. SCF-Type E3 Ligases Are Involved in Pulmonary Fibrosis

EMT of alveolar epithelial cells to α-SMA-positive myofibroblasts and promotion of fibroblast proliferation participate in development of pulmonary fibrosis. In cell proliferation, cyclin-dependent kinases (CDKs) play important roles as the cell cycle engine. CDK inhibitors (CKIs) such as p27^Kip1^ (p27) and p21^Cip1^ (p21), participate in regulation of the growth and differentiation of both epithelial and interstitial cells in various tissues [107]. A complex-type RING E3 SCF-Skp2 degrades p27 and p21 as a typical E3 ligase for them via proteasome [108,109]. SCF-type E3 ligases include Cullin1, which acts as a scaffold, Rbx1/Roc1 that have RING finger domains, Skp1 that binds to F-box, and F-box proteins that participate in binding to specific substrates [108,110]. There are at least 70 F-box proteins in humans, such as Skp2, Fbw7 (Fbxw7), and β-TrCP. Skp2 mainly targets negative regulators of the cell cycle, such as p27, p21, p57, RBL2, and TOB1 for ubiquitin-mediated proteasomal degradation and participates in cell proliferation [111].

In the progressive kidney obstruction mouse model, Skp2 deficiency suppresses renal fibrosis and the progression of renal fibrosis is partially recovered by Skp2/p27-double deficiency [112,113]. In the model, Skp2-mediated degradation of p27 and p21 is required for progression of kidney fibrosis by proliferation of renal tubular epithelial cells and interstitial cells, respectively [114]. In deficient mice, α-SMA-positive myofibroblasts also decrease which suggests that Skp2 is involved in renal cell proliferation and epithelial-to-myofibroblast transition during the progression of renal fibrosis.

Similarly, in the bleomycin-induced pulmonary fibrosis model, accumulation of both collagen A1 and fibronectin and the score of pulmonary fibrosis are reduced by Skp2-deficiency [87]. Moreover, p27-positive vimentin-positive mesenchymal fibroblasts which decrease in bleomycin-treated mice, are recovered by Skp2 deficiency, which suggests that both proliferation and population of mesenchymal fibroblasts are increased by Skp2-deficiency (Table 1). These results suggest that Skp2-mediated p27 degradation is involved in the progression of pulmonary fibrosis by promotion of either mesenchymal fibroblast proliferation, EMT, or both (Figure 3). Additionally, Skp2 inhibitor (SZ-P1-41) treatment partially inhibits progression of bleomycin induced fibrosis, which suggests that a Skp2 inhibitor may be applicable to IPF therapy [87].

SCF-Fbw7 targets several growth promoting proteins such as c-Myc, c-Myb, Notch, and GATA2/3, whereas SCF-Skp2 targets several anti-proliferate proteins such as p27 and p21 [111]. Wang et al. found that Fbw7 deficiency inhibits stress-induced senescence of type II alveolar epithelial stem cells and pulmonary fibrosis in the bleomycin-induced mouse model. Stability of telomere protection protein 1 (TPP1) increases in Fbw7-depleted A549 cells, which suggests that Fbw7 participates in TPP1 degradation as an E3 ligase (Figure 3). Not only lentivirus-mediated depletion of Fbw7, but also TELEODIN, which specifically inhibits binding of Fbw7 to TPP1, suppresses telomere shortening, senescence of type II alveolar epithelial stem cells. Thereby alveolar epithelial cell number is restored and pulmonary fibrosis is suppressed in the bleomycin-model [88] (Table 1). Therefore, an Fbw7 inhibitor such as TELEODIN may be applicable to IPF therapy.

## 8. Conclusions

IPF is a progressive and fatal disease with 2.5–3.5 years being the median survival time after diagnosis. Although incomplete, understanding of IPF has recently progressed. The criteria for IPF diagnosis were recently revised to emphasize the significance of the radiological UIP pattern and multidisciplinary discussion between pulmonologists, radiologists, and pathologists in the diagnostic process. Although there is currently no curative treatment for IPF, promising medications, including pirfenidone and nintedanib, have been introduced. They reduce the decline in lung function in patients with IPF, but have a small impact on survival and quality of life.

Advances in understanding fibrogenic processes are likely to yield more effective therapies that give longer survival and better quality of life. The current concepts of IPF pathogenesis focus on epithelial cell damage, the TGF-β system, and epithelial–mesenchymal transition. It would be helpful to consider other processes in the pathogenesis of pulmonary fibrosis, such as the ubiquitin-proteasome pathway and endothelial cells. Many targeted therapies are currently in early-phase clinical trials. Signaling related to the ubiquitin-proteasome pathway may be another therapeutic target. Further studies with multifaceted and broader approaches will contribute to better understanding of the molecular pathogenesis of pulmonary fibrosis.

## Figures and Tables

**Figure 1 ijms-22-06107-f001:**
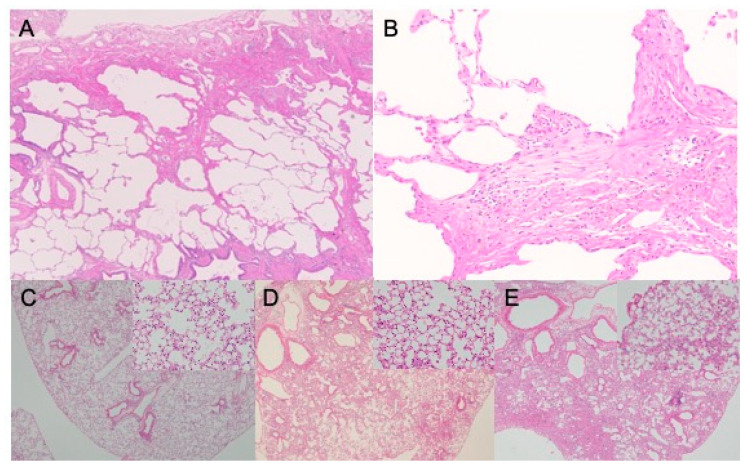
Representative optical microscopy images of fibrotic lungs. Hematoxylin-eosin stain for human idiopathic pulmonary fibrosis (**A**,**B**) and bleomycin-treated mouse lungs at day 7 (**C**), day 21 (**D**), and day 28 (**E**). The images of human idiopathic pulmonary fibrosis show usual interstitial pneumonia pattern, architectural destruction, heterogeneous patchy distribution of fibrosis with scarring and honeycomb changes, and scattered fibroblastic foci.

**Figure 2 ijms-22-06107-f002:**
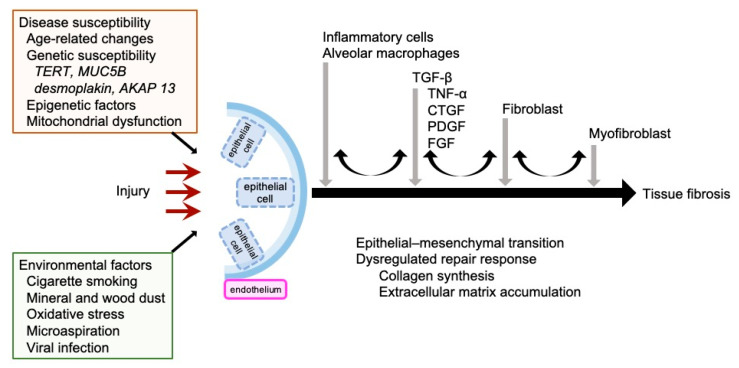
Current concepts of IPF pathogenesis. *AKAP 13*, A-kinase anchoring protein 13; CTGF, connective tissue growth factor; FGF, fibroblast growth factor; *MUC5B*, mucin 5B; PDGF, platelet-derived growth factor; *TERT*, telomerase reverse transcriptase; TNF-α, tumor necrosis factor-α.

**Figure 3 ijms-22-06107-f003:**
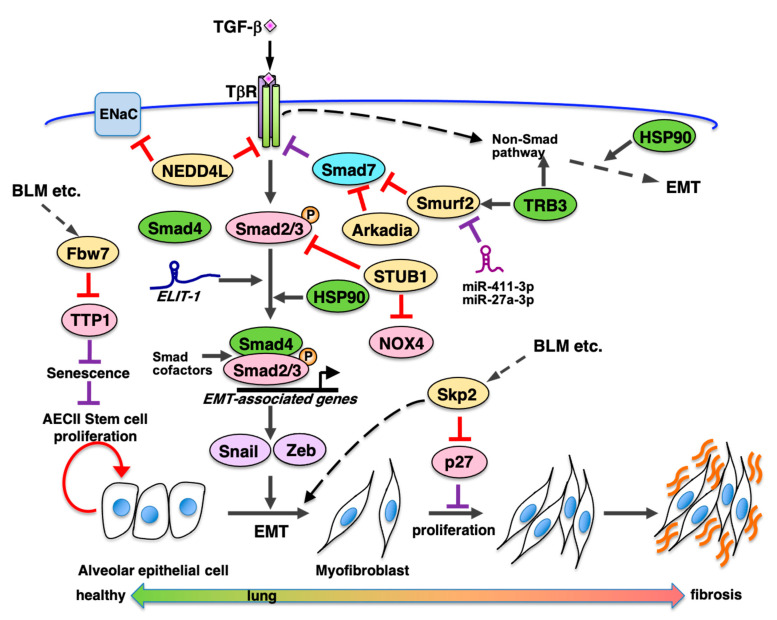
Possible modulators of fibrosis via regulating TGF-β signaling and cell proliferation. TGF-β signaling promotes induction of epithelial–mesenchymal transition (EMT) and expression of fibrosis-associated genes via TGF-β-Smad-dependent canonical pathway and non-Smad pathway. In the canonical pathway, TGF-β binds to and activates its receptor complex and promotes phosphorylation of Smad2/3. Then activated Smad2/3 forms a complex with Smad4 and the complex binds to the promoter regions of target genes associated with EMT and fibrosis. Inhibitory Smad such as Smad7 and Smad cofactors to regulate the TGF-β-Smad pathway. MiRNA such as miR-21 inhibits translation of Smad7 as a positive regulator of TGF-β signaling. Several lncRNAs participate in controlling TGF-β-mediated EMT and/or fibrosis. A lncRNA *ELIT-1* which is induced by TGF-β-Smad pathway binds to Smad3 to facilitate recruitment to promoters of Smad target genes promoting EMT as a Smad cofactor. Moreover, several E3 ubiquitin ligases are involved in promotion in pulmonary fibrosis. Skp2 which promotes p27 degradation is involved in promotion of EMT and mesenchymal fibroblast proliferation. Fbw7 which participates in TPP1 degradation promotes alveolar epithelial stem cell senescence and pulmonary fibrosis. Abbreviations and details of each molecule are indicated in the text.

**Table 1 ijms-22-06107-t001:** E3 ligases involved in promotion and suppression of pulmonary fibrosis.

E3 Ligase	Type	Target in Fibrosis	Effect on Fibrosis	Function of the E3 Ligase on Pulmonary Fibrosis	Ref.
Smurf2	HECT	Smad7	Promotive	Promotion of TGFβ-signaling by degradation of Smad7	[81,82]
NEDD4L	HECT	ENaC, TβR ^1^	Suppressive	Promotion of degradation of ENaC and TβR in type II alveolar epithelial cell	[83,84]
Arkadia	RING	Smad7	Promotive	Promotion of TGFβ-signaling by degradation of Smad7 methylated by Set9	[85]
STUB1	U-box	Smad3, NOX4	Suppressive	Inhibition of myofibroblast differentiation by degradation of NOX4 and Smad3	[86]
Skp2	SCF	p21, p27	Promotive	Involvement in mesenchymal fibroblasts proliferation by degradation of p27	[87]
Fbw7	SCF	TPP1 ^2^	Promotive	Promotion of type II alveolar epithelial cell senescence by degradation of TPP1	[88]

^1^ TβR TGF-β receptor; ^2^ TPP1, telomere protection protein 1

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
