# Peer review of "Molecular Pathogenesis of Pulmonary Fibrosis, with Focus on Pathways Related to TGF-β and the Ubiquitin-Proteasome Pathway"

_ijms, 2021, doi:10.3390/ijms22116107_

Round 1
Reviewer 1 Report
The authors present a manuscript entitled “Molecular pathogenesis of pulmonary fibrosis.” My comments are outlined below.
Overall, I think the authors do a very nice job giving an overview of molecular pathways that are potentially involved with pulmonary fibrosis, as well as reviewing a little bit of the clinical features of pulmonary fibrosis, including its radiology and histopathology manifestations. Overall, I think the manuscript is well written, though I do think that the abstract could be a little better written. I have a few suggestions as outlined below.
Suggestions:
- In section 2.1, the authors state that involvement of the inflammatory pathway is regarded to be mild and to have only a partial in the onset and progression of disease. I would recommend the language here to be changed slightly. There are numerous studies demonstrating that the inflammatory components of pulmonary fibrosis (both cellular and molecular inflammation) are actually quite substantial and dramatic, but as the authors state accurately, it does seem clear that anti-inflammatory or immunosuppressive therapies are not efficacious in IPF. I would just suggest the authors revise this statement a little bit that evidence of cellular and molecular inflammation can be quite dramatic in some patients with IPF, but the data would suggest that there is not a causal role for inflammation in many patients with IPF.
- Also in section 2.1, the authors discussed broadly the concept of aberrant and dysregulated wound healing, and also used the term impaired wound healing. I would just make a slight distinction here, and not equate aberrant with impaired. For example, if a patient has a mutation in one of the telomere biology genes and the epithelium is unable to repair itself due to a genetic mutation, the subsequent repair response may not necessarily be aberrant or dysregulated, it may just be that the repair response is impaired due to the epithelium being able to sustain itself due to abnormal telomere biology. I would suggest that the authors just not equate the term aberrant with the term impaired.
- In section 3, when the authors discuss bleomycin administration as an animal model, the authors accurately state that fibrotic changes in response to bleomycin resolve spontaneously after 28 days. The authors also accurately state this seems to be a major limitation of the bleomycin model, and that no animal model to date has been able to reproduce the progressive fibrotic phenotype observed in patients with IPF. I would suggest the authors just emphasize this point a little more in this section regarding animal models.
- Overall the manuscript I think is well written, but does seem to have a major focus towards the end of the manuscript on TGF-beta, epithelial-mesenchymal transition, and the ubiquitin-proteasome pathway. The manuscript does not mention much at all many other molecular pathways that potentially could be involved in pulmonary fibrosis, including pathways related to the innate immune system, adaptive immune system, oxidative stress, coagulation disturbances, and the role of genetic mutations. Consequently, I would suggest the title of the manuscript could convey this focus on TGF-beta, EMT, ubiquitin-proteasome pathway. The authors can decide what is best in this regard, but perhaps something such as "molecular pathogenesis of pulmonary fibrosis, with focus on pathways related to TGF-beta and the ubiquitin-proteasome pathway.” I think this would be a little more informative to the reader when they read the manuscript.
- As I mentioned above, I think overall the manuscript is well written, but I think the abstract in my judgment is not as well written, with some grammatical errors in the abstract which do not seem to be present throughout the manuscript. Consequently, rewriting the abstract and rewording a few of the sentences and sentence structure would be helpful to the reader, and creating a little more broadly applicable final sentence.
Author Response
Response to Reviewer 1
Point 1: In section 2.1, the authors state that involvement of the inflammatory pathway is regarded to be mild and to have only a partial in the onset and progression of disease. I would recommend the language here to be changed slightly. There are numerous studies demonstrating that the inflammatory components of pulmonary fibrosis (both cellular and molecular inflammation) are actually quite substantial and dramatic, but as the authors state accurately, it does seem clear that anti-inflammatory or immunosuppressive therapies are not efficacious in IPF. I would just suggest the authors revise this statement a little bit that evidence of cellular and molecular inflammation can be quite dramatic in some patients with IPF, but the data would suggest that there is not a causal role for inflammation in many patients with IPF.
Response 1: We appreciate your positive comments and suggestion. We have rewritten this section and added that numerous studies have demonstrated that inflammatory components are actually quite substantial and dramatic in the pathogenesis of IPF.
Point 2: In section 2.1, the authors discussed broadly the concept of aberrant and dysregulated wound healing, and also used the term impaired wound healing. I would just make a slight distinction here, and not equate aberrant with impaired. I would suggest that the authors just not equate the term aberrant with the term impaired.
Response 2: We agree and have changed "aberrant" throughout the manuscript.
Point 3: In section 3, when the authors discuss bleomycin administration as an animal model, the authors accurately state that fibrotic changes in response to bleomycin resolve spontaneously after 28 days. The authors also accurately state this seems to be a major limitation of the bleomycin model, and that no animal model to date has been able to reproduce the progressive fibrotic phenotype observed in patients with IPF. I would suggest the authors just emphasize this point a little more in this section regarding animal models.
Response 3: Thank you for your suggestion. We have rewritten this section to focus on the limitation of the bleomycin model.
Point 4: The manuscript does seem to have a major focus on TGF-beta, epithelial-mesenchymal transition, and the ubiquitin-proteasome pathway. The title of the manuscript focusing on them would be a little more informative to the reader.
Response 4: Thank you for your kind suggestion. We have changed the title accordingly.
Point 5: I think overall the manuscript is well written, but I think the abstract is not as well written with some grammatical errors. Consequently, rewriting the abstract and rewording a few of the sentences and sentence structure would be helpful to the reader, and creating a little more broadly applicable final sentence.
Response 5: Thank you for your suggestion. We have rewritten the abstract and the revised manuscript has been edited and proofread by two native-English-speaking science editors from Edanz Group Ltd.
Reviewer 2 Report
The manuscript discuss the new advances in IPF pathogenesis. The topic is interesting, however a major English editing is necessary, specifically for the abstract and the introduction.
Authors should perform a more thorough review the manuscript as sentences are not well written and appear disconnected within each section.
The animal model section spends emphasis on a specific model (44) with no clear explanation. The section should be rewritten.
Additional evidence regarding the inflammasome should be included (https://www.frontiersin.org/articles/10.3389/fimmu.2021.642855/full).
The section on ubiquitin-proteasome pathways could be improved by including the recent evidence of HSP90 inhibitors (https://www.mdpi.com/1422-0067/21/15/5286 ; https://erj.ersjournals.com/content/49/2/1602152
A major revision is required.
Author Response
Response to Reviewer 2
Point 1: The topic is interesting, however, a major English editing is necessary, specifically for the abstract and the introduction. Authors should perform a more thorough review the manuscript as sentences are not well written.
Response 1: Thank you for your suggestion. We have rewritten the abstract and the revised manuscript has been edited and proofread by two native-English-speaking science editors from Edanz Group Ltd.
Point 2: The animal model section spends emphasis on a specific model with no clear explanation. The section should be rewritten.
Response 2: We have rewritten this section focusing on the advantages and the limitation of the bleomycin model.
Point 3: Additional evidence regarding the inflammasome should be included.
Response 3: We have added text regarding the inflammasome with appropriate references according to the reviewer’s suggestion.
Point 4: The section on ubiquitin-proteasome pathways could be improved by including the recent evidence of HSP90 inhibitors.
Response 4: We have added text regarding the recent evidence of HSP90 inhibitors with appropriate references and we have revised Figure 3.
Round 2
Reviewer 2 Report
The authors have addresses revewer comments. The manuscript has been improved. No mroe comments